# Effects of Sediments Phosphorus Inactivation on the Life Strategies of *Myriophyllum spicatum*: Implications for Lake Restoration

Zhenmei Lin [1], Chen Zhong [2], Guolong Yu [3], Yishu Fu [4], Baohua Guan [1,5], Zhengwen Liu [1,4,5,*] and Jinlei Yu [5,*]

[1] Sino-Danish Centre for Education and Research (SDC), University of Chinese Academy of Sciences, Beijing 100190, China; linzhenmei19@mails.ucas.ac.cn (Z.L.); bhguan@niglas.ac.cn (B.G.)

[2] Department of Science and Technology, China Gezhouba Group Water Operation Co., Ltd., Wuhan 430030, China; chenzhong@cggc.cn

[3] Fuyang Gezhouba Guozhen Water Environment Treatment Investment and Construction Co., Ltd., Fuyang 236000, China; yvguolong123@163.com

[4] Department of Ecology and Institute of Hydrobiology, Jinan University, Guangzhou 510630, China; elainfys@163.com

[5] State Key Laboratory of Lake Science and Environment, Nanjing Institute of Geography and Limnology, Chinese Academy of Sciences, Nanjing 210008, China

* Correspondence: zliu@niglas.ac.cn (Z.L.); jlyu@niglas.ac.cn (J.Y.); Tel.: +86-025-86882013 (Z.L.); +86-025-8688-2113 (J.Y.)

**Abstract:** Eutrophication often results in the loss of submerged vegetation in shallow lakes and turns the lake to be a turbid state. Recovery of submerged macrophytes is the key in the restoration of shallow eutrophic lakes to create a clear water state. However, internal loading control was considered as the critical process for the recovery of submerged macrophytes in shallow lakes after the external nutrient reduction. Phoslock® (Lanthanum modified bentonite) is a useful passivation material in controlling the internal loadings (release of phosphorus from the sediments), which was applied to restore the eutrophic lakes. However, the effects of Phoslock® on the growth and life strategies of submerged macrophytes are less focused so far. In the present study, we studied the responses in the growth and morphological characteristics of *Myriophyllum spicatum* to the addition of Phoslock® to the sediments. Our results showed that the addition of Phoslock® significantly decreased the contents of bioavailable forms of phosphorus in the sediments, such as redox-sensitive phosphorus bound to Fe and Mn compounds (BD–P), phosphorus bound to aluminum (Al–P) and organic phosphorus (Org–P). However, the concentration of the non-bioavailable forms of phosphorus in the sediments, such as calcium bound phosphorus (Ca–P), increased significantly in the Phoslock® treatments compared with the controls. At the end of the experiments, the total biomass, aboveground biomass and relative growth rate (RGR) of *M. spicatum* decreased significantly in the Phoslock® mesocosms compared with the controls. In contrast, the wet root biomass, root–shoot biomass ratio, root numbers and root length of *M. spicatum* were significantly higher in the Phoslock® treatments than that in the controls. Our results indicated that the growth of *M. spicatum* was suppressed by the addition of Phoslock®, and thus the biomass was decreased; however, the increase of root biomass might be beneficial to the inhibition of phosphorus release and resuspension of sediments and to the restoration of the lake ecosystem.

**Keywords:** internal loading control; Phoslock®; submerged macrophytes; *Myriophyllum spicatum*; lake restoration

## 1. Introduction

There are two alternative states of equilibrium in shallow lakes, the turbid states dominated by high biomass of phytoplankton and the clear-water state dominated by high coverage of submerged macrophytes [1–3]. The anthropogenic eutrophication leads to the

increase in phytoplankton biomass and the loss of submerged macrophytes [4–6]. However, submerged macrophytes have many positive feedbacks on lake abiotic and biotic variables; for example, they can absorb nutrients from both the water and the sediments, inhibit the resuspension of the sediments, offer refuge for the large-bodied zooplankton escaping from the predation by fish [1,7,8] and produce allelochemicals that suppress phytoplankton growth [9,10]. Therefore, the biomass of phytoplankton is generally low in submerged macrophyte-dominated lakes [4,11]. The recovery of submerged macrophyte communities has become crucial in stabilizing a clear-water state of shallow lakes [12–14]. The key to restoring the lake vegetation community is to reduce the loading of nutrients since excessive nutrients will lead to eutrophication and affect the growth and community composition of aquatic plants [15,16]. However, the natural recovery of the submerged macrophytes community shows a delayed response to the external loading reductions [17–19]. This may be due to the rich nutrients in the sediments (internal loading), which can support the high biomass of phytoplankton and induce a low light availability suppressing the growth of submerged macrophytes [19–22]. In the last two decades, transplantation of submerged macrophytes was conducted to enhance the establishment of submerged macrophyte communities in subtropical and tropical shallow lakes, especially in China [12,23–25].

Usually, in order to enhance the light availability for submerged macrophytes and reduce excess nutrients in the water, phosphorus inactivation methods are commonly used in combating lake eutrophication [26–28]. For the removal of internal phosphorus loading in the water and sediments, there are chemical (e.g., in situ chemical inactivation), physical (e.g., sediment dredging and aeration) and biological methods (e.g., biomanipulation) [29–31]. Phosphorus inactivation materials, such as salts of iron, aluminum, calcium or lanthanum, and modified clays and soils were continuously improved and applied in controlling internal phosphorus loadings [32–34]. For instance, Li et al. [30] conducted calcium silicate hydrates (CSH) to control the release of phosphorus in the lake sediments; they found that CSH had a significant effect on the immobilization of phosphorus in sediment. Phosphorus inactivation methods using aluminum compounds also induced the reduction in bioavailable phosphorus in the sediments [33,35]. Phoslock® (Lanthanum modified bentonite), developed by the Australian CSIRO in the 1990s, is a passivation material to control the release of phosphorus from the sediments, and its addition is effective in case studies with the reduction of algal blooms caused by internal phosphorus release [34,36,37]. The concentrations of nutrients in the water, the nutrient release rate from the sediments to the water and the concentration of total porewater soluble reactive phosphorus in the surface layers all decrease markedly after the application of lanthanum modified bentonite [29,32,38]. However, the reduction in nutrient availability in both the water column and the sediments may affect the life strategies of submerged macrophytes.

The growth and community dynamics of submerged macrophytes are influenced by both biotic and abiotic factors, such as light availability, water temperature, nutrient concentrations in both the water and the sediments, and water flow [7,39,40]. The life history-related parameters of aquatic plants will change with the changing circumstances to avoid or resist in order to reduce the potentially negative effects from the adverse environmental conditions [41–44]. For instance, the reduced availability of nutrients in the sediments leads to the increase in biomass allocation to the root system resulting in a higher root to shoot ratio [45,46]. Additionally, plants decreased the allocation of N and P to stem with increasing water total nitrogen concentration [47]. Moreover, due to the species-specific life history strategies of plants, different environmental factors, such as the ambient nutrient contents, have different effects on the plants [48].

In this study, we conducted a mesocosm experiment to test the effects of Phoslock® addition to the sediments on the growth and life strategy of *Myriophyllum spicatum* with two treatments, control (without the addition of Phoslock® to the sediments) and Phoslock® treatment (with the addition of Phoslock® to the sediments). We hypothesized that the growth of *M. spicatum* was suppressed as a result of the reduction in the concentrations of bioavailable phosphorus in the sediments after the addition of Phoslock® to the sedi-

ments. We also hypothesized that *M. spicatum* would adjust its resource allocation (e.g., increasing the belowground biomass) to adapt to the decrease in bioavailable phosphorus in the sediments.

## 2. Materials and Methods

### 2.1. Analyzed Species

*Myriophyllum spicatum* is a submerged macrophyte species that is widely distributed in Europe, Asia and North Africa [49]. It is an invasive species that needed to be controlled in North America [50,51]. *M. spicatum* is used to adsorb different pollutants in sewage water. For example, alginate encapsulated with *M. spicatum* has a high adsorption capacity of Pb (II) [49]. Moreover, *M. spicatum* is a typical rooted submerged macrophyte species that can absorb nutrients via both shoot and root [45,52,53], which is commonly used in restoring eutrophic shallow lakes [23,25].

### 2.2. Experiment Set-Up

The experiment was conducted at Jinan University, Guangzhou, China, in December 2018 and lasted for 28 days. The experiment consisted of 10 transparent buckets (with an inner diameter of 24 cm and a height of 22 cm), which were then divided into two treatments, the control (without the addition of Phoslock® to the sediments) and the Phoslock® (with the addition of Phoslock® to the sediments) group, with five replicates for each treatment. Each bucket was filled with 10 cm sediments collected from Minghu Lake in Jinan University. The sediments were mixed well and then filtered through a 0.5 cm meshed sieve to remove large particles. The initial concentrations of total nitrogen (TN) and total phosphorus (TP) of the sediments were 3.8 mg·g$^{-1}$ and 3.3 mg·g$^{-1}$, respectively. Phoslock® were randomly added to the sediments in 5 of 10 buckets, and the sediments were fully stirred and evenly mixed with Phoslock®. The amount of Phoslock® was calculated according to the ratio of La$^{3+}$: PO$_4$$^{3-}$ = 1:1, where PO$_4$$^{3-}$ was the content of mobile P fractions (NH$_4$Cl-P, BD-P and Org-P) in the sediments [54,55], and La$^{3+}$ was the content of lanthanum in Phoslock® (5% lanthanum in Phoslock®) produced by Australia's Phoslock Environmental Technologies LTD. Thereafter, ten individuals of *M. spicatum* (with a mean height of 13.0 ± 0.5 cm and a mean biomass of 1.0 ± 0.2 g) were planted in each of the buckets. Finally, all the buckets were suspended in a tank (120 cm in diameter and 120 cm in height) at a depth of 50 cm.

### 2.3. Sample Collection and Measurements

Sediment samples were collected at the beginning (before the addition of Phoslock®) and the end of the experiment. In each sampling event, a 50 mL syringe (2 cm in diameter, 8 cm in length, with the front end cut off) was used to collect the surface sediments (upper 5 cm) at three different locations in each bucket. The samples from the same bucket were then mixed well and stored at −20 °C in the refrigerator for further analysis. Subsamples of the sediments were used to analyze the concentrations of different forms of phosphorus. The content of TP in the sediments was determined according to Aspila et al. [56]. Different forms of phosphorus, loosely sorbed P (NH$_4$Cl-P), redox-sensitive P forms bound to Fe and Mn compounds (BD–P), organic P (Org–P), P bound to aluminum (Al–P), P bound to calcium (Ca–P) and residual P (Res-P) [57], were analyzed in line with SMT (Standard Measurements Testing Program of European Union) protocol [58].

At the end of the experiment, all *M. spicatum* were sampled and cleaned with tap water in the laboratory. The biomass, shoot length, shoot diameter, leaf number, root length and root number of each plant were measured. The total, aboveground, and belowground biomass of *M. spicatum*, was calculated. The shoot length, shoot diameter and root length were measured by a vernier caliper. The plants were dried at 80 °C for 24 h and grounded into powder to determine the content of TN and TP in the whole plant. The ratio of root–

shoot (R:S) biomass was calculated with belowground biomass/aboveground biomass. The relative growth rate (*RGR*) of *M. spicatum* was calculated using Equation (1) [59]:

$$RGR\left(g\cdot g^{-1}\cdot day^{-1}\right) = \ln\left(W_f/W_i\right)/days \tag{1}$$

where $W_f$(g) and $W_i$(g) are the final and initial total biomass of *M. spicatum* in each bucket, respectively.

### 2.4. Data Analysis

All results were presented with the means and standard deviations of 5 buckets. The differences in the content of different forms of phosphorus in the sediments, and the biomass, relative growth rate and other plant traits of *M. spicatum* between the two treatments were compared using one-way ANOVA by an LSD post hoc test in SPSS software 24.0 (IBM, New York, NY, USA). All data were tested for normality and homogeneity of variances in SPSS before analysis. The *p* values less than 0.05 were considered statistically significant. All the figures were plotted in the Origin 9.1 software.

## 3. Results

### 3.1. The Concentrations of Different Forms of Phosphorus in the Sediments

At the end of the experiment, the concentrations of BD–P ($F_{1,8}$ = 5.8, *p* = 0.043), Al–P ($F_{1,8}$ = 6.1, *p* = 0.039) and Org–P ($F_{1,8}$ = 5.5, *p* = 0.048) in the Phoslock® addition treatments were all significantly lower than those in the control group (Figure 1). In contrast, the concentrations of Ca–P in the Phoslock® treatment was substantially higher than in the controls ($F_{1,8}$ = 17.3, *p* = 0.003; Figure 1). No significant differences were found for the concentrations of both NH₄Cl-P ($F_{1,8}$ = 0.06, *p* = 0.82) and Res-P ($F_{1,8}$ = 3.27, *p* = 0.11) between the two treatments (Figure 1).

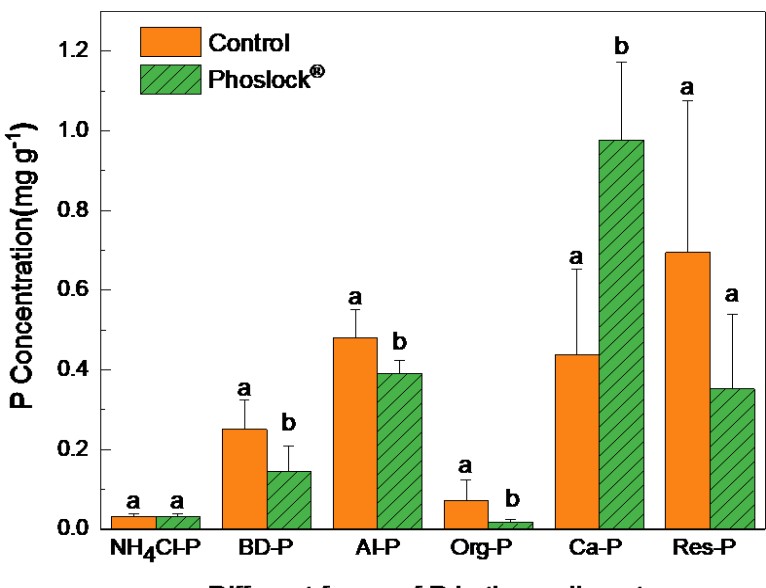

**Figure 1.** The contents of different forms of phosphorus in the sediments in the control (without Phoslock® addition) and Phoslock® (with Phoslock® addition) treatment at the end of the experiment. Different letters indicate a significant difference between the two treatments at *p* < 0.05. Error bars represent the standard deviations (SD).

### 3.2. Growth and Biomass Allocation of M. spicatum

The total biomass ($F_{1,8}$ = 9.3, *p* = 0.016; Figure 2a) and relative growth rate ($F_{1,8}$ = 15.4, *p* = 0.004; Figure 2b) of *M. spicatum* were both significantly suppressed by the addition of Phoslock® to the sediments compared with the controls. The aboveground biomass of

the plants in the Phoslock® group was significantly lower than in the controls ($F_{1,8}$ = 13.5, $p$ = 0.006; Figure 2c); however, the belowground biomass was pronouncedly higher than the controls ($F_{1,8}$ = 9.1, $p$ = 0.017; Figure 2c). Therefore, the root to shoot ratio of *M. spicatum* in the Phoslock® treatment was significantly higher than that in the control group ($F_{1,8}$ = 43.1, $p$ < 0.0001; Figure 2d).

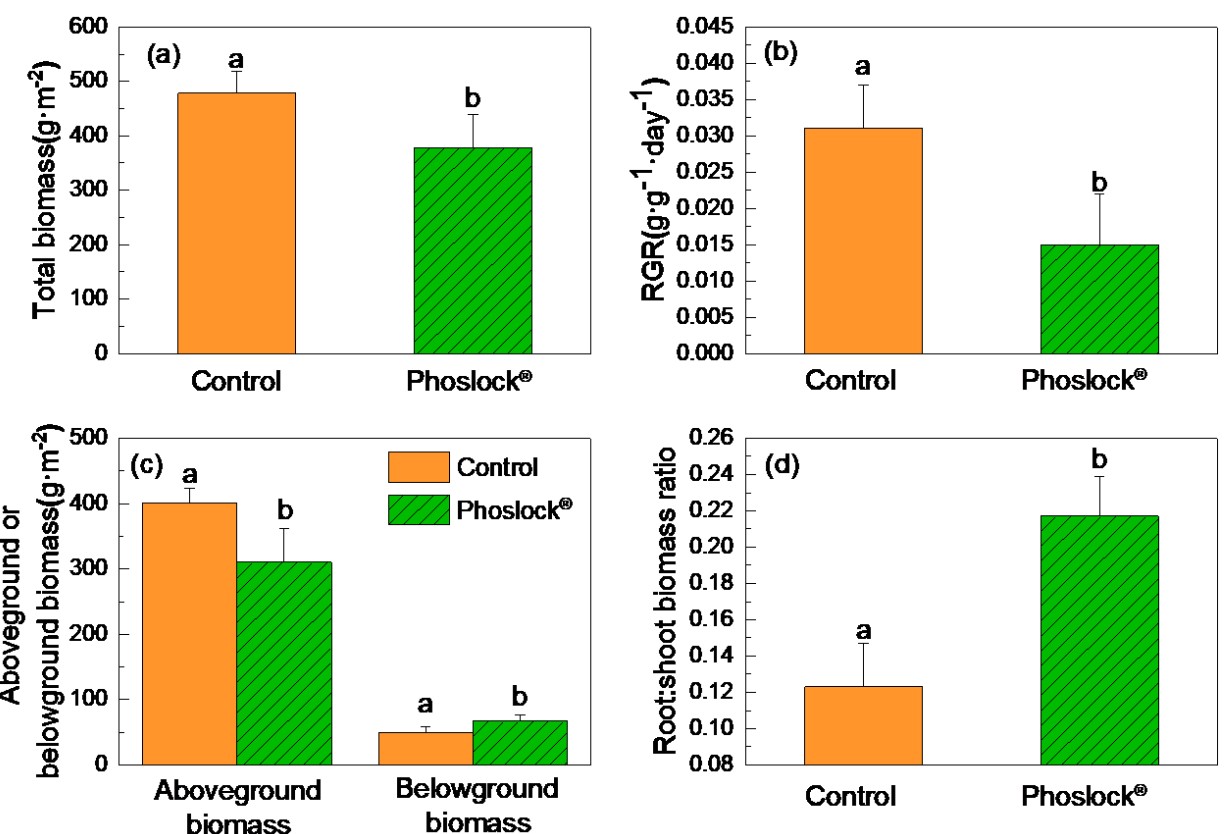

**Figure 2.** The differences in the total biomass (**a**), relative growth rate (**b**), aboveground and belowground biomass (**c**) and root: shoot biomass ratio (**d**) of *Myriolhyllum spicatum* in the control (without Phoslock® addition) and Phoslock® (with Phoslock® addition) treatments at the end of experiment. Different letters indicate a significant difference among treatments at $p$ < 0.05. Error bars represent the standard deviations (SD).

### 3.3. Morphological Traits of M. spicatum

At the end of the experiment, the mean shoot diameter of *M. spicatum* in the Phoslock® treatment did not differ significantly with that of the controls ($F_{1,8}$ = 0.4, $p$ = 0.53; Figure 3a). The mean shoot length ($F_{1,8}$ = 14.3, $p$ = 0.005; Figure 3b) and leaf number ($F_{1,8}$ = 50.9, $p$ < 0.0001; Figure 3c) of *M. spicatum* were significantly suppressed by the addition of Phoslock® to the sediments compared with the control group. However, the mean length ($F_{1,8}$ = 105.4, $p$ < 0.0001; Figure 3d) and number ($F_{1,8}$ = 50.7, $p$ < 0.0001; Figure 3d) of the roots in the Phoslock® addition buckets were both significantly higher than in the controls.

### 3.4. Contents of Nitrogen and Phosphorus in the Plant

The TP contents of *M. spicatum* in the Phoslock® addition mesocosms were significantly lower than that in the controls ($F_{1,8}$ = 14.6, $p$ = 0.005; Figure 4), while the TN content of the plants did not differ substantially between the two treatments ($F_{1,8}$ = 0.09, $p$ = 0.77; Figure 4).

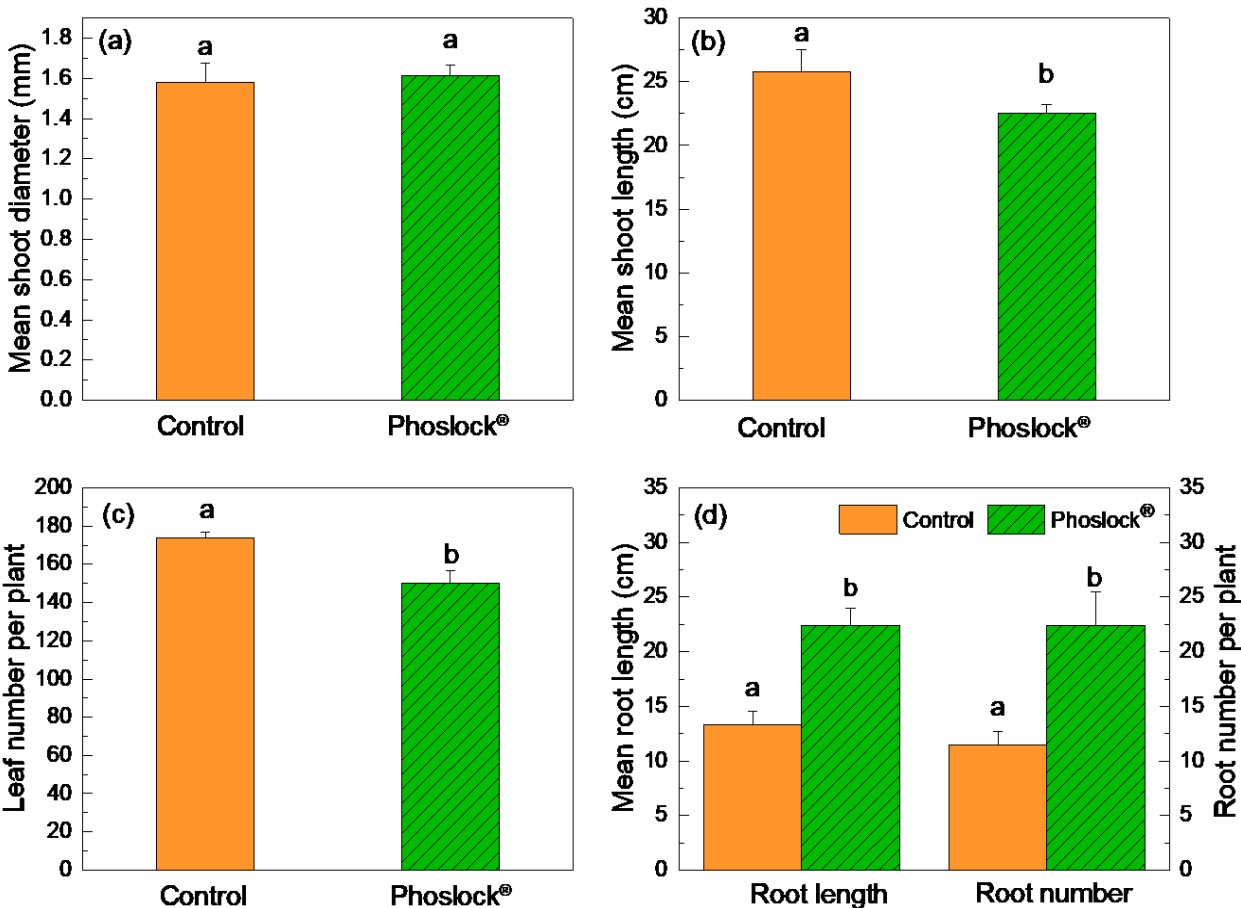

**Figure 3.** Morphological parameters of *Myriolhyllum spicatum* included mean shoot diameter (**a**), mean shoot length (**b**), leaf number per plant (**c**) and root parameters (**d**) in both the control (without Phoslock® addition) and Phoslock® (with Phoslock® addition) treatments at the end of experiment. Different letters indicate a significant difference between the two treatments at $p < 0.05$. Error bars represent the standard deviations (SD).

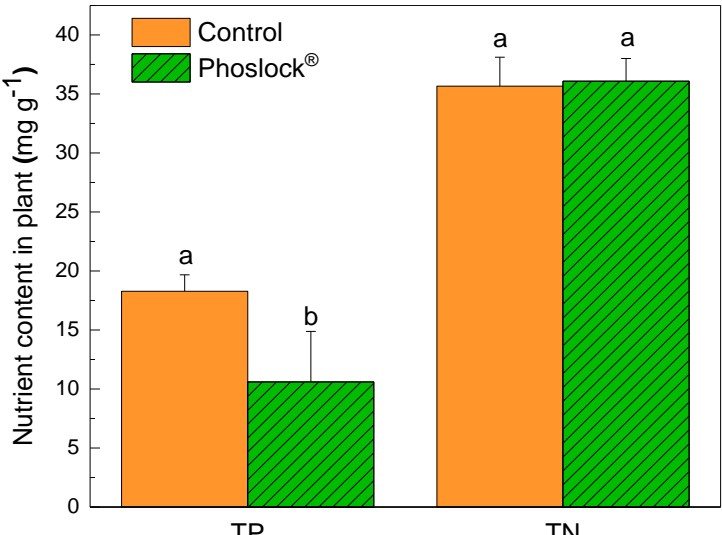

**Figure 4.** The contents of phosphorus and nitrogen in *Myriolhyllum spicatum* in the control (without Phoslock® addition) and Phoslock® addition treatments, respectively. Different letters indicate a significant difference between the two treatments at $p < 0.05$. Error bars represent the standard deviations (SD).

## 4. Discussion

We found that the addition of Phoslock® to the sediment reduced the concentration of bioavailable forms of phosphorus in the sediments. Therefore, the growth (RGR) of *M. spicatum* was suppressed by the decreasing availability of phosphorus in the sediment. Moreover, the life strategies of *M. spicatum* turned to accumulate more belowground biomass in the Phoslock® addition treatment than the control group.

In our study, the change in bioavailable phosphorus in the sediments impacted the resource allocation strategy of *M. spicatum* in the Phoslock® addition group. Resource allocation is one of the central contents of modern ecological life history theory study [60], and plants adjust resource allocation to optimize resource capture [61,62]. The trade-off in resource allocation is mainly through the allocation of different nutrients within specific organs and the allocation of the same nutrient between organs [47,63]. The allocation of plant biomass between root and shoot (the ratio of belowground biomass to aboveground biomass) is an important indicator for the evaluation of plant life strategy change with environmental factors. Plants absorb carbon mainly through their aboveground parts for photosynthesis, while phosphorus and other nutrients are assimilated mainly through their belowground parts [64]. Thornley [65] argued that the allocation of biomass between roots and shoots was affected by the concentration of nutrients in the water, such as carbon and phosphorus. The increase in carbon concentration in overlying water will increase the root biomass [65,66] and promote the absorption of nutrients from sediments. However, the increase in phosphorus and other nutrients in the sediments enhances the biomass of both plant shoot and leaf, thereby stimulating the rates of both photosynthesis and growth of the plants [61,65].

*M. spicatum* can absorb nutrients through both the shoots and leaves [53]; however, its strong root system is considered to be the main organ for absorbing phosphorus from the sediments [67,68]. Previous studies have shown that the growth rate of *M. spicatum* increases with the enrichment of nutrient concentration in the sediments [68,69]. The biomass allocation between roots and shoots was also affected by the nutrient content of sediments; for instance, the root– : shoot ratio of submersed macrophyte *M. spicatum* decreased with the increase of nutrient content in the sediments [69]. However, in the present study, sediment phosphorus inactivation (Phoslock®) significantly reduced the total biomass of *M. spicatum* while markedly increased the root– : shoot ratio. This may attribute to the reduction in bioavailable phosphorus in the sediments after the addition of Phoslock®, which may suppress the growth of *M. spicatum*. Life strategies change in aquatic plant species, such as the root– : shoot ratio, is common in adapting to soil resources variations [45], leading to the change in the biomass allocation between aboveground and belowground of the plants. We found that the total phosphorus (TP) content of *M. spicatum* in the Phoslock® addition group was significantly lower than that in the control group, which gives further evidence on the effects of reduced bioavailable phosphorus in sediments on the growth of macrophytes. In addition, the increase in the number of roots and the root length of *M. spicatum* in the Phoslock® treatment compared to the controls may also be attributed to the reduction in bioavailable phosphorus in the sediment. Our results were in line with Xie et al. [70], who found that the reduction in nutrient availability (e.g., nitrogen and phosphorus) in the sediments significantly increased the root length of *M. spicatum*.

The critical mechanism of submerged macrophytes in determining the lake ecosystem states is inhibition of both the nutrient release and resuspension of sediments [1,71]. The biomass of submerged macrophytes is an important factor in affecting sediment resuspension because the high biomass of macrophytes will reduce the sediment resuspension rate [72,73]. In our study, phosphorus inactivation by the addition of Phoslock® reduces the total biomass of *M. spicatum*, which may increase the risk of sediment resuspension and nutrients released from the sediments. However, Marin-Diaz et al. [74] showed that the effects of macrophytes on sediment resuspension were positively highly related to their underground biomass. In the present study, the number and biomass of roots of *M. spicatum*

were significantly enhanced by the addition of Phoslock®. The amounts of phosphorus release from the sediments are in correlation with the oxidation of sediments by the roots of submerged macrophytes. Oxygen release to the sediments by the roots of macrophytes can improve the redox potential in the surface sediments, thus enhancing the retention capacity for phosphorus in sediments and inhibiting the release rate of phosphorus to the overlying water [75,76]. Previous studies showed that the oxygen concentration in the sediments was the critical factor in affecting the release of phosphorus from the sediments to the overlying water [77,78]. The increased dissolved oxygen of the sediments can reduce the release rate of phosphorus from the sediments to the water, thereby reducing the internal phosphorus loading and speeding up the recovery of eutrophic lakes. Our results indicated that phosphorus inactivation can increase the belowground biomass, the number and length of roots of *M. spicatum*, and may further enhance the phosphorus retention capacity of sediments. The interacting effects of phosphorus inactivation and submerged macrophytes on the internal phosphorus loading from sediments need to be further studied in situ experiments.

## 5. Conclusions

We found that the addition of Phoslock® changed the contents of different forms of phosphorus in sediments. The contents of bioavailable phosphorus, such as redox-sensitive phosphorus bound to Fe and Mn compounds (BD–P), phosphorus bound to aluminum (Al–P) and organic phosphorus (Org–P), decreased significantly, while the non-bioavailable phosphorus such as calcium bound phosphorus (Ca–P) increased significantly. As a result, the total biomass and aboveground biomass of *M. spicatum* decreased significantly, while the root biomass, root– : shoot biomass ratio, root numbers and root length increased significantly. Our results suggested that though the relative growth rate of *M. spicatum* was suppressed, the root biomass was enhanced due to the reduction in bioavailable phosphorus content in the sediments, which might be beneficial to the inhibition of phosphorus release from the sediments and resuspension of sediments.

**Author Contributions:** Conceptualization, Z.L. (Zhengwen Liu); methodology, Z.L. (Zhengwen Liu); validation, Y.F., C.Z. and G.Y.; formal analysis, Z.L. (Zhenmei Lin); investigation, Y.F.; resources, Y.F.; data curation, Y.F. and Z.L. (Zhenmei Lin); writing—original draft preparation, Z.L. (Zhenmei Lin), B.G., Z.L. (Zhenmei Lin) and J.Y.; writing—review and editing, Z.L. (Zhenmei Lin), B.G., Z.L. (Zhengwen Liu) and J.Y.; visualization, Z.L. (Zhengwen Liu) and J.Y.; supervision, Z.L. (Zhengwen Liu) and J.Y.; funding acquisition, Z.L. (Zhengwen Liu). All authors have read and agreed to the published version of the manuscript.

**Funding:** This research was funded by the Chinese National Key Research and Development Project (2017YFA0605201) and the National Science Foundation of China (41877415).

**Institutional Review Board Statement:** Not applicable.

**Informed Consent Statement:** Not applicable.

**Data Availability Statement:** Data are presented in the text.

**Acknowledgments:** We thank Manli Xia, Yehui Huang, Mingming Zhu and Hongye He for field and laboratory support.

**Conflicts of Interest:** The authors declare no conflict of interest.

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
