# Peer review of "Effects of Sediments Phosphorus Inactivation on the Life Strategies of Myriophyllum spicatum: Implications for Lake Restoration"

_water, doi:10.3390/w13152112_

Round 1
Reviewer 1 Report
The Ms 'Effects of sediments phosphorus inactivation on the life strategies of Myriophyllum spicatum: implications for lake restoration' by Zhenmei Lin et al. presents a simple and short (1 month) mesocosm experiment concerning the effect of phosphorus inactivation on aquatic plants and discuss the few ecological implications on lake recovery.
Even if the research content is not novel, the ms shows results that could be interesting for management and recovery of eutrophic lakes. In my opinion it would be much more newsworthy about long term effects.
Good part of references sounds dated and there are many self-citations. Introduction and discussion need to be supported by recent and 'wide' literature.
The research design seems appropriate.
Results need to be improved and modified. First of all is mandatory to analyse the variance of INTRA-treatment (control and treatment) and demonstrate that it is significantly low than the variace INTER-treatment. Furthermore to use ANOVA, data have to be normally distributed. The relative growth rate (RGR) of M. spicatum not seems to be interesting as the authors measure the growth only at the beginning and the end of the experiment and they not give elements about different stage of development during the time in the two treatments.
In discussion the authors speculate on 'the internal phosphorus loading from sediments' but they not measure the effect of submerged plants on that process.
Author Response
Reviewer #1: The Ms 'Effects of sediments phosphorus inactivation on the life strategies of Myriophyllum spicatum: implications for lake restoration' by Zhenmei Lin et al. presents a simple and short (1 month) mesocosm experiment concerning the effect of phosphorus inactivation on aquatic plants and discuss the few ecological implications on lake recovery.
Even if the research content is not novel, the ms shows results that could be interesting for management and recovery of eutrophic lakes. In my opinion it would be much more newsworthy about long term effects.
- Good part of references sounds dated and there are many self-citations. Introduction and discussion need to be supported by recent and 'wide' literature.
Our reply: Thanks. More recent and ‘wide’ literatures have been added to introduction and discussion now.
The research design seems appropriate.
- Results need to be improved and modified. First of all is mandatory to analyse the variance of INTRA-treatment (control and treatment) and demonstrate that it is significantly low than the variance INTER-treatment. Furthermore to use ANOVA, data have to be normally distributed.
Our reply: Thanks. We added “All data were tested for normality and homogeneity of variances in SPSS before analysis.” in line 180-181.
- The relative growth rate (RGR) of spicatumnot seems to be interesting as the authors measure the growth only at the beginning and the end of the experiment and they not give elements about different stage of development during the time in the two treatments.
Our reply: Thanks for your inspiring comments which we will follow in the future study. Relative growth rate (RGR) is the most widely-used method to compare growth rates among species or genotypes (Paul-Victor et al., 2010). Although different stage of development during the time is not available, we could clearly know the average change of the biomass of M. spicatum in the two treatment groups by calculating RGR, so as to judge that the Phoslock® addition inhibited the growth of M. spicatum.
- In discussion the authors speculate on 'the internal phosphorus loading from sediments' but they not measure the effect of submerged plants on that process.
Our reply: In the present study, we were intending to determine the effects of the application of PhosLock® in the sediment on the life strategies of the submerged macrophyte M. spicatum. We have discussed the responses of M. spicatum to the reduction of bioavailable phosphorus both in the water and the sediments in lines 507-543. In addition, the implications of our results, especially the life strategies changes of the plant, on shallow lake restoration were also discussed in lines 544-582. The effects of submerged plants on the control of internal phosphorus loading were not studied in our experiments.
Reviewer 2 Report
Overall, the work is very well done. It is well structured and, importantly, has very clear illustrations and is well written. My main comments are more in the nature of minor suggestions, although I think they should be taken into account before likely publication:
The introduction is very limited and does not introduce the reader to the essence of the problem under analysis. It should be more elaborated, especially the part from lines 60-72. Moreover, the text on lines 73-75 should be moved to Methods and there I suggest inserting the section Analyzed species.
Section 2.3 Data lacks a description of the tests used to analyses the assumptions of ANOVA, i.e., how the homogeneity of variance was tested, the homogeneity of the groups analyzed, as well as whether the condition about normality of distribution was met.
Suggested literature:
GENERAL (about factors and species)
Response of aquatic plants to abiotic factors: a review
G Bornette, S Puijalon
Aquatic sciences 73 (1), 1-14
Plant resistance to mechanical stress: evidence of an avoidance–tolerance trade‐off
S Puijalon, TJ Bouma, CJ Douady, J van Groenendael, NPR Anten, ...
New Phytologist 191 (4), 1141-1149
The interaction between wetland nutrient content and plant quality controls aquatic plant decomposition
C Grasset, LH Levrey, C Delolme, F Arthaud, G Bornette
Wetlands ecology and management 25 (2), 211-219
Flow field downstream of individual aquatic plants—Experiments in a natural river with Potamogeton crispus L. and Myriophyllum spicatum L.
Ł Przyborowski, AM Łoboda, RJ Bialik, K Västilä
Hydrological Processes 33 (9), 1324-1337
SPECIFIC (about Phoslock)
Performance of aquatic weed-Waste Myriophyllum spicatum immobilized in alginate beads for the removal of Pb (II)
JV Milojković, ZR Lopičić, IP Anastopoulos, JT Petrović, SZ Milićević, ...
Journal of environmental management 232, 97-109
Assessing the responses of aquatic macrophytes to the application of a lanthanum modified bentonite clay, at Loch Flemington, Scotland, UK
IDM Gunn, S Meis, SC Maberly, BM Spears
Hydrobiologia 737 (1), 309-320
Effect of calcium silicate hydrates coupled with Myriophyllum spicatum on phosphorus release and immobilization in shallow lake sediment
C Li, H Yu, S Tabassum, L Li, Y Mu, D Wu…
Chemical Engineering Journal 331, 462-470.
Author Response
Reviewer #2: Overall, the work is very well done. It is well structured and, importantly, has very clear illustrations and is well written. My main comments are more in the nature of minor suggestions, although I think they should be taken into account before likely publication:
- The introduction is very limited and does not introduce the reader to the essence of the problem under analysis. It should be more elaborated, especially the part from lines 60-72. Moreover, the text on lines 73-75 should be moved to Methods and there I suggest inserting the section Analyzed species.
Our reply: Thank you very much for your suggestions, and we have made further supplement to the introduction part in the Ms.
- Section 2.3 Data lacks a description of the tests used to analyses the assumptions of ANOVA, i.e., how the homogeneity of variance was tested, the homogeneity of the groups analyzed, as well as whether the condition about normality of distribution was met.
Our reply: Thanks. We added the description of the tests in lines 443-444.
- Suggested literature:
GENERAL (about factors and species)
Response of aquatic plants to abiotic factors: a review G Bornette, S Puijalon Aquatic sciences 73 (1), 1-14
Plant resistance to mechanical stress: evidence of an avoidance–tolerance trade‐off
S Puijalon, TJ Bouma, CJ Douady, J van Groenendael, NPR Anten, ...New Phytologist 191 (4), 1141-1149
The interaction between wetland nutrient content and plant quality controls aquatic plant decomposition C Grasset, LH Levrey, C Delolme, F Arthaud, G Bornette Wetlands ecology and management 25 (2), 211-219
Flow field downstream of individual aquatic plants—Experiments in a natural river with Potamogeton crispus L. and Myriophyllum spicatum L. Ł Przyborowski, AM Łoboda, RJ Bialik, K Västilä. Hydrological Processes 33 (9), 1324-1337
SPECIFIC (about Phoslock)
Performance of aquatic weed-Waste Myriophyllum spicatum immobilized in alginate beads for the removal of Pb (II) JV Milojković, ZR Lopičić, IP Anastopoulos, JT Petrović, SZ Milićević, ... Journal of environmental management 232, 97-109
Assessing the responses of aquatic macrophytes to the application of a lanthanum modified bentonite clay, at Loch Flemington, Scotland, UK IDM Gunn, S Meis, SC Maberly, BM Spears Hydrobiologia 737 (1), 309-320
Effect of calcium silicate hydrates coupled with Myriophyllum spicatum on phosphorus release and immobilization in shallow lake sediment C Li, H Yu, S Tabassum, L Li, Y Mu, D Wu…Chemical Engineering Journal 331, 462-470.
Our reply: Thanks for the useful literatures. All the suggested references have been cited now.
Round 2
Reviewer 2 Report
The paper has been improved since the first time was reviewed. I think that now can be published in MDPI Water Journal.